# Product Quality during the Storage of Foods with Insects as an Ingredient: Impact of Particle Size, Antioxidant, Oil Content and Salt Content

**DOI:** 10.3390/foods9060791

**Published:** 2020-06-16

**Authors:** Karin Wendin, Lennart Mårtensson, Henric Djerf, Maud Langton

**Affiliations:** 1Faculty of Natural Sciences, Kristianstad University, SE-291 88 Kristianstad, Sweden; lennart.martensson@hkr.se (L.M.); henric.djerf@hkr.se (H.D.); 2Department of Food Science, University of Copenhagen, DK-1958 Frederiksberg C, Denmark; 3Department of Molecular Sciences, SLU, Swedish University of Agricultural Sciences, SE-750 07 Uppsala, Sweden; maud.langton@slu.se

**Keywords:** yellow mealworm, *Tenebrio molitor*, insects, sensory, processed, model system, shelf life

## Abstract

To increase the acceptability of insects as food in Western culture, it is essential to develop attractive, high-quality food products. Higher acceptability of insect-based food has been shown if the insects are “invisible”. Mincing or chopping the insect material could be a first processing step to reduce the visibility of the insects. In this work, we processed yellow mealworms by using traditional food techniques: chopping, mixing and heat treatment in a retort. The results show that all factors in the experimental design (particle size, oil content, salt content and antioxidant) influenced the products to a larger extent than the storage time. The results, measured by sensory analysis, TBAR values (Thiobarbituric acid reactive substances), colourimetry and viscosity, show clearly that the food products packaged in TRC (Tetra recart cartons) 200 packages and processed in a retort stayed stable during a storage time of 6 months at room temperature.

## 1. Introduction

Globally, insects are popular as human food in many countries [1,2]. Edible insects, such as yellow mealworms, may be a sustainable source of protein and fat since they are efficient feed converters with a high nutritional value [1,2]. Yellow mealworms have been shown to contain many of the essential amino acids, such as histidine, isoleucine, leucine, lysine, methionine + cysteine, phenylalanine + tyrosine, threonine and valine [3]. There are numerous studies on insects as human food, however very few focus on their taste and flavour or show their potential to become tasty foods [4].

The development of attractive, high-quality food products is essential if the consumption of insect-based food products is to increase in Western culture. Tan et al. [5] pointed out that insect-based food products are unlikely to be accepted if they do not meet the consumers’ expectations. The use of “insect-flour” containing non-visible insects had a significantly higher acceptance than the use of whole and visible insects in studies in both Sweden [6] and Denmark [7]. Mincing or chopping the insect material could be a first processing step to reduce the visibility of the insects. Another way of reducing visibility is to use insects as an ingredient; for example, Azzollini et al. [8] added ground yellow mealworm larvae blended with wheat flour to produce extruded cereal snacks. Extraction of the insect proteins may also be an alternative to the use of whole insects [9].

There is a need to understand the mechanisms affecting the sensory properties and oxidation stability of insects when used as food. The primary attributes of the sensory quality of a food product depend on the product type and may be due to appearance (colour, size, shape), odour, flavour, taste and texture [10]. In a previous study, it was shown that the salt content and particle size had a high impact on the sensory attributes of mealworm samples in a model food system [11]. Salt level and particle size are therefore important factors to evaluate in the development of food products based on insects. Salt usually affects protein aggregation, solubility and rheological properties [12]. Soy protein isolate was found to form a gel upon heating, which is common for many proteins. Chen et al. [12] also found that adding salt increased the storage modulus of soy protein isolate gels. However, in a previous study where extracted mealworm protein gels were analysed, the addition of 1.0% NaCl was found to affect the storage modulus G’, which decreased. This was explained by the change in solubility [3]. In contrast to many other proteins, the storage modulus G’ decreased during heating in extracted mealworm protein [3]. Thus, mealworm protein could exhibit different behaviours compared to many other proteins. The addition of an antioxidant agent, extract of rosemary, seemed to affect the colour, rancidity and separation; however, further studies are needed [11]. For this purpose, sensory characteristics can by evaluated by a trained sensory panel and results can then be related to instrumental analysis of viscosity and colour. Rancidity is often measured using measurements of Thiobarbituric acid reactive substances (TBAR), i.e., the degradation products of lipids, which can be detected through the use of a reagent consisting of thiobarbituric acid [13,14] Another common way to measure TBAR values is through the use of a commercially available Lipid Peroxidation (Malondialdehyde (MDA)) Assay Kit [15].

Food products need to be processed and packaged in order to maintain sensory characteristics and safety, and to increase the shelf life of the food. Dobermann [16] suggests that clear processing and storage methodologies should be established for insects as food. Heat treatment is often used to prevent microbial growth and to produce safe products. A retort is commonly used for the preservation of many food products [17]. The packaging is also of importance and must include barriers to oxygen and light to prevent the development of rancidity during storage [18].

Product browning occurs in minced mealworm, possibly due to enzymes, according to Tonneijck-Srpová et al. [19]. They showed that inactivation of the enzymes responsible for browning could be achieved by blanching and/or high-pressure processing at 400 and 500 MPa. Nevertheless, this also resulted in a decrease in the texturizing properties of the minced mealworms, such as lower values for fracture stress and Young’s modulus, particularly when blanching was used.

In this study, a food model system was produced according to an experimental design [20]. In order to create model food products similar to real food, oil, water, salt and antioxidant were added to finely grated or coarsely chopped mealworms. These additions can affect the mealworm products during storage due to stability problems that may occur. The model products can be compared to food products such as paté or purée, or Swedish meatballs. Patés typically have a fat content around 25% but may vary between 10% and 30% [21] and Swedish meatballs also vary between 10% and 30%, with commercial products containing 17% [22]. We used sunflower oil in the model products. Both yellow mealworms and sunflower oil contain unsaturated fatty acids, which make the products susceptible to oxidation and a rancid off-flavour can therefore occur during storage. Rosemary extract was used as an antioxidant and combined with packaging with highly protective barriers, which was used to prevent oxidation and increase the shelf life of the model products based on insects.

## 2. Aim

Food model products based on yellow mealworms were put into packages and processed in a retort to evaluate the impact of particle size, oil/water ratio, salt content and amount of antioxidant on the sensory properties, viscosity, rancidity and colour during storage, compared to freshly produced products.

## 3. Materials and Methods

### 3.1. Material

The fresh Yellow Mealworms (*Tenebrio molitor*) were reared on a small scale in Sweden. The breeding of the mealworms took place in plastic boxes, about 300 mm × 300 mm × 100 mm in size and kept at an ambient temperature (approximately 22 °C). Humidity was approximately 50%. The time from egg to mealworm was 12 weeks. The mealworms were fed on oat bran and carrot. Sunflower oil (Farm, Swedish Fine Rice and Food AB, Huddinge, Sweden), salt (NaCl, Nordfalks, Sweden) and an antioxidant based on rosemary (Duralox Oxidation Management Blend NM-45, HT, NS, product code: 62-103-03) were also used. The products were packaged in 200 mL Tetra Recart cartons (TRC 200). The nutrient composition was analysed by ALS Scandinavia AB, Danderyd, Sweden (Table 1).

The composition of the mealworm is such that it could provide a useful contribution to the intakes of zinc and iron, especially zinc, since the zinc content of mealworm is high at 34 mg/kg. Thus, the intake of this type of food could be a useful addition to the daily intake. The recommended daily intakes of zinc and iron are both around 10 mg, depending on age and gender [23,24]. The product also contains omega 3 and omega 6 fatty acids, with high levels of omega 6, and therefore could contribute to the intake of this essential fatty acid. The level of cadmium is low (Table 1) and therefore only a high consumption of the product would exceed the toxic level.

### 3.2. Experimental Design

Four different samples were produced according to a factorial design (Figure 1). The recipes for each of the samples were the following:

**1.** 4 kg meal worm, 1 kg oil, 3 kg water, 80 g salt, 0.36 g rosemary extract, resulting in an oil content around 12.5%.

**2.** 4 kg meal worm, 3 kg oil, 1 kg water, 80 g salt, 11.2 g rosemary extract, resulting in an oil content around 37.5%.

**3.** 4 kg mealworm, 1 kg water, 3 kg oil, 20 g salt, 0.36 g rosemary extract, resulting in an oil content around 37.5%.

**4.** 4 kg mealworm, 3 kg water, 1 kg oil, 20 g salt, 11.2 g rosemary extract, resulting in an oil content around 12.5%.

### 3.3. Production

A total of 16 kg of mealworms were provided, allowing a maximum of 45 packages of each recipe to be prepared. The manufacturing process for the products will now be described.

Frozen mealworms were blanched for 2–3 min and then chopped by hand or grated using a kitchen blender before the packages were filled. The ingredients for the liquid were mixed and then added to the chopped or grated mealworms. Each package was filled to the top manually before sealing.

A standard sterilisation process was selected for the heat treatment of the products at 122 °C while monitoring the F-value of the product. The four samples were processed in a JBT Retort: AR092-T JBT Mobile Test Unit (Belgium). Production, packaging and processing took place at Tetra Pak, Lund Sweden.

The four different products were run 2 by 2 in the retort, thus allowing all products to be processed on the same day. Samples with a similar oil content were put into the same retort since differences had been found when monitoring product-specific F-values during a pre-trial. Recipes 1 and 4 were therefore run together, as were recipes 2 and 3. Data from the heating process are presented in Table 2. Each package was labelled with a date of withdrawal for shelf life analysis, which was 6 months after production day. All sample batches were tested using microbiological analysis: 5 packages incubated at 30 °C for 7 days and 4 packages incubated at 55 °C for 7 days. The food products were evaluated on Plate Count Agar after package incubation. Results for all samples were <1 Colony Forming Unit (CFU)/10 microlitre. After microbiological approval, the samples were released for use in the project.

### 3.4. Storage

Packaged samples were stored at an ambient temperature of approximately 21 °C. The samples were retrieved 7 times, first directly after preparation and then each month for 6 months, and then put into a freezer (Vestfrost solutions VTS 098, Esbjerg, Denmark) and kept at −80 °C until analysis. All samples were thawed before analysis and analysed at the same time. The samples were labelled as shown in Table 3.

### 3.5. Sensory Analysis

The sensory panel was selected and trained according to ISO (International Standard Organisation) standard 8586-2:2008 [26]. The panel consisted of eight assessors who were trained to perform a quantitative descriptive analysis. During training, reference samples were presented, and their sensory attributes were defined (Table 4). The panel was further trained in how to rate attribute intensities on a numerical scale from 0 to 100. The assessments were then performed in triplicate, in a randomised order. The panel was instructed to use water and neutral wafers to cleanse the palate and neutralise the senses. The assessors were informed not to swallow the samples but to spit them out after assessment. The software Eye Question, Netherlands, was used for sensory data collection. Each assessor signed up for participation after being informed about the products and the terms of participation: voluntary participation, freedom to leave the test without giving a reason and the right to decline to answer specific questions.

### 3.6. Instrumental Measurements

Instrumental measurements (colour, viscosity and rancidity) were performed on all 28 samples (four samples at 7 different time points) plus replicates, see Section 3.4.

Colour measurements were performed using a spectrophotometer (Konica Minolta CM-700d) by measuring L*-, a*-, and b*-values, where L* corresponds to the lightness, a* to the red-green scale (red is positive and green negative) and b* to the yellow-blue scale (yellow is positive and blue is negative) [27].

Rheology is the study of the deformation and flow of materials and can describe the properties of materials, ranging from liquids to elastic solids. One common rheological method is the measurement of viscosity. In this study, the viscosities of the samples were measured using a rotational viscometer (Brookfield DVII+, AMETEK Brookfield Inc., Middleborough, MA, USA) equipped with an S1 spider.

The measurements of rancidity were performed using Thiobarbituric acid reactive substances (TBAR). For these analyses, a Lipid Peroxidation (MDA, Malonaldehyde) Assay Kit (catalogue number MAK085, Sigma-Aldrich, St. Louis, MO, USA) was used for colorimetric assay at λ = 532 nm. The frozen samples stored at −80 °C were thawed before analysis. Since the samples were a mixture of shell parts and soft parts, a stainless-steel garlic press was used to obtain a more homogeneous material. 1.2 mg of the material was weighed into a microcentrifuge vial (three replicates per sample) and centrifuged at 16,000 *g* for 30 min. 10 μL of the supernatant was transferred into a microcentrifuge vial and gently mixed with 500 μL of 42 mM sulfuric acid. 125 μL of phosphotungstic acid solution was then added and mixed by vortexing. The vial was then incubated at room temperature for 5 min before centrifuging at 13,000 *g* for 3 min. The pellet was resuspended on ice with water/Butylated hydroxy toluene (100 μL/2 μL) solution and the volume was adjusted to 200 μL with water. A 0.1 M MDA-solution was prepared for the standard calibration curve. This was further diluted into a 0.2 mM MDA standard solution and 0, 2, 4, 6, 8 and 10 μL of the 0.2 mM MDA standard solution were added to separate microcentrifuge tubes. The volume in each vial was adjusted by the addition of water to 200 μL. This generated 0 (blank), 4, 8, 12, 16 and 20 nmole standards. To form the MBA-TBA adduct, 600 μL of a TBA solution was added to each vial containing standard and sample followed by incubation at 95 °C for 60 min. After this, the vials were cooled in an ice bath for 10 min. 200 μL from each vial was pipetted into 96-well plates for analysis in a plate reader for colorimetric assay at λ = 532 nm.

### 3.7. Statistical Analyses

Data were analysed by calculating mean values and standard deviations. The data were further subjected to two-way analysis of variance (ANOVA) with samples and measurements as fixed effects. Significant differences (*p* < 0.05) between samples were calculated via the Tukey’s Post Hoc pairwise comparison test. Results were correlated by Pearson correlation. Regression analysis was performed where the design variables were independent and the measurements were dependent factors (IBM SPSS, version 26). Finally, principal component analysis (PCA; Panel Check V 1.4.2, Nofima, Norway) was performed to give an overview of the results.

## 4. Results

All the insect food samples were analysed during the same week. Figure 2 shows the sensory attributes of (a) fresh samples and (b) after 6 months of storage. In both fresh and stored samples, the particle size in the attribute “appearance of roughness” was detected, where samples 3 and 4 were perceived as having a statistically significantly higher roughness and also contained the coarsely chopped mealworms. The two samples with higher salt content, samples 1 and 2, were also perceived as significantly saltier. Samples 2 and 3 were perceived as having a significantly higher degree of oiliness, which became more apparent after 6 months of storage. These two samples had a higher oil content. During storage, the odours and flavours of popcorn and roasted sesame seed decreased significantly in all four samples. The decrease was largest at the beginning of the storage.

Figure 3 shows the sensory attributes of the four samples during storage. The same pattern was almost maintained during the 6 months of storage. Figure 3 also shows similar patterns for samples 1 and 2 (with high values for glossy, brownish-grey and grittiness) that were also different from samples 3 and 4 (with high values for roughness).

Table 5 shows the mean values for viscosity, colour and TBAR for the samples when fresh and during six months of storage. The values of viscosity can be seen as an indication of the samples being very inhomogeneous. There were also very few significant differences. Colour measured by L-value seems higher for smaller particles (samples 1 and 2) in Table 5, and all samples differed significantly at the sample level. The difference between particle size was significant for b*-values. The yellow appearance was significantly lower for two samples (1 and 2; Figure 2). Table 5 also shows that there was very little change during storage for up to 6 months in the packages at room temperature, which is supported by the low change in TBAR values over time.

Figure 4 shows a PCA plot of the mean values over time for the four samples, showing 97.6% of the variance in the resulting data. The four samples are well distributed in each of the four quadrants. Along the first principal component, showing 75.0% of the variance, samples 3 and 4 (coarsely chopped mealworm and low salt content) are plotted close to roughness and opposite samples 1 and 2 (finely grated mealworm and high salt content). The second principal component, representing 22.6% of the variance, was TBAR, with samples 1 and 4 (with low oil content) showing opposite glossy appearance, oiliness and viscosity, which are instead close to samples 2 and 3 (with high oil content). Overall, the results show that the design had a clear impact on the measured parameters.

The results from the PCA are supported by the regression analysis, which showed that all the design parameters had a higher impact than storage time. Oil and salt contents had the largest impact on most attributes, but particle size and rosemary also had significant impacts, mainly on the appearance attributes. The resulting sensorial, rheological, colour and TBAR values were influenced by the design factors to a higher extent than storage time. The only parameters being significantly influenced by the storage time were the odour and flavour attributes of oat, popcorn and roasted sesame seed.

The Pearson correlations also support the PCA results. Table 6 shows that viscosity was negatively correlated to TBAR values. Tsaltiness was positively correlated to Abrownish-grey and Tsourness and negatively correlated to Tsweetness, Ayellow and Aroughness. Further, there was a high correlation between the odour and flavour attributes of popcorn and roasted sesame seed.

## 5. Discussion

As early as 1975, Meyer-Rochow pointed out that insects could be used as a protein source for humans [28], however even though insects seem to be a sustainable alternative compared to livestock [29], this is not enough for consumers. Food products need to be made with appealing sensory characteristics in order for consumers to buy and eat them; in addition, they must maintain their quality during storage. Sensory determination of shelf life in food products is more or less routine in product development [30]. In this work, we processed yellow mealworms by using traditional food techniques such as chopping, mixing and heat treatment in a retort. The procedures were performed in order to reduce the visibility of the mealworms and to create a model product advised by the results of a focus group in a previous study [6]. The group suggested using ground insects as an ingredient in meat dishes such as sausages and hamburgers. Sausages typically have a fat content between 25% and 30%. Our products contained 12.5% or 37.5% fat, which can be regarded as levels of fat that are commonly used in food products today. The ingredients were included in an experimental design and varied according to input from the focus group. The number of product samples had to be kept low since the total number of samples was large due to the analysis during storage. The study included all 28 samples consisting of the four samples from the experimental design being taken out for analysis at seven different time points during the storage time. A full factorial design would give a larger number of samples from the experimental design [20,31]; however, the number of time points for analysis would then have to decrease.

It was clear from the regression analysis that all the factors in the experimental design influenced the products to a larger extent than the storage time. The results show that the particle size has an impact on the visual appearance of the product. We found that smaller particles, that is finely grated mealworm in samples 1 and 2, had higher values of brownish-grey appearance independent of being fresh or during storage time. However, the samples with larger particles, samples 3 and 4 (coarsely ground mealworm), were perceived as having high values of roughness. The results in this study are, with a few exceptions, in agreement with the results in a previous study [11], where particle size had a large impact on the evaluated parameters, such as brownish-grey appearance. Our results show that the brownish-grey colour was influenced by the particle size and did not change during storage. This is in line with Tonneijick-Srpová et al. [19], who found that browning of minced mealworms could be reduced by blanching in air-tight packaging.

The large change in oil content from 12.5% to 37.5% affected the glossy appearance and the mouthfeel of oiliness. The samples with a higher oil content were perceived to be both more glossy and more oily than the others.

The two samples with higher salt contents were perceived as saltier; here, we cannot detect the effect of particle size as this coincides with the salt levels. Larger liquid volumes in the product could result in a faster release of salt in the mouth, which we were not able to detect in this experiment.

Rosemary level had an impact, mainly on the appearance attributes. This was indicated in a pilot study by Wendin et al. [11], where rosemary seemed to influence both colour and separation. The rosemary extract was added as an antioxidant and was assumed to reduce possible oxidation [32,33]. Nevertheless, no rancidity was perceived by the panel, nor was it detected in the TBAR values. In our previous work, we used the addition of 0%, 0.1% and 0.3% of rosemary extract, where rosemary had a positive impact in preventing rancidity and, as already mentioned, changes in colour and phase separation in samples stored in vacuum-sealed packages in the fridge (5 °C) for 15 days. All four products in this study were very stable during the 6-month storage time, indicating an effective process and well-functioning packaging material.

During storage there was, however, a significant decrease in some of the volatile odour compounds. The sensory panel clearly detected a drop in the odours and flavours of popcorn, oat and roasted sesame seed. This was supported by the regression analysis, which showed that storage time had a significant influence on these sensory attributes. The decrease could probably be related to changes in the chemical composition of volatiles during storage [34].

The results, measured by sensory analysis, TBAR values, colourimetry and viscosity, clearly show that the food products packaged in TRC 200 packages and processed in a retort stayed stable during a storage time of 6 months at room temperature, with the exception of a decrease in the odour and flavour attributes of popcorn and roasted sesame seed.

## 6. Conclusions

It can be concluded that the measured sensory properties of viscosity, rancidity and colour were impacted mainly by the design parameters of salt and oil content, and to a smaller extent by particle size and the addition of an antioxidant. Storage time of the processed and packaged insect products influenced only the odour and flavour attributes of popcorn and roasted sesame seed. This means that products based on insects are stable during storage when processed in a retort and packaged in TRC packages.

## Figures and Tables

**Figure 1 foods-09-00791-f001:**
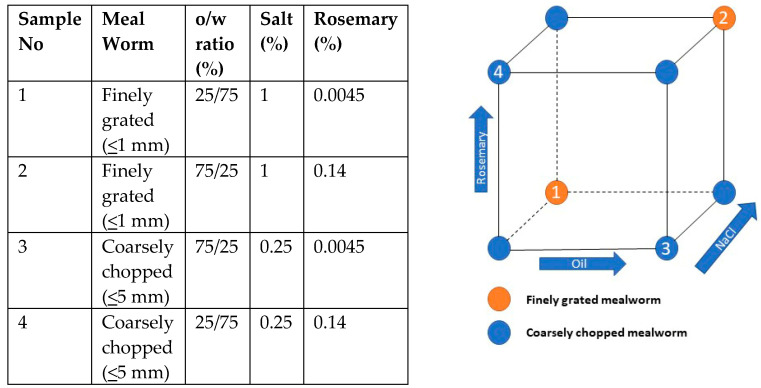
Illustration of the experimental design. To the **left** an overview of the design and to the **right** an illustration of the design oil content ~12.5% or ~37.5%, NaCl content ~0.25% or ~1%, and rosemary content ~0.0045% or ~0.14%.

**Figure 2 foods-09-00791-f002:**
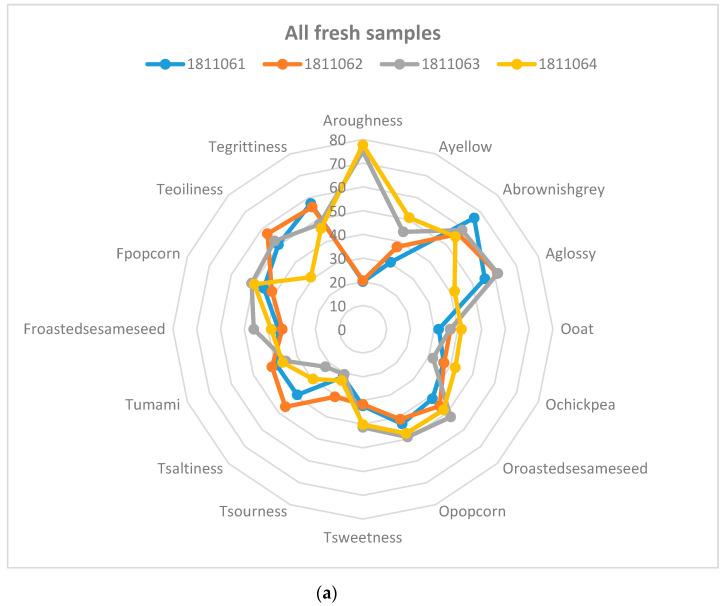
Spider diagram of sensory attributes of (**a**) fresh samples and (**b**) after 6 months of storage.

**Figure 3 foods-09-00791-f003:**
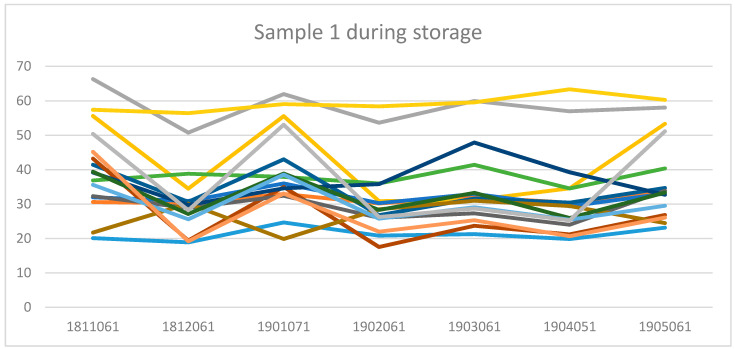
The perceived sensory attributes of all four samples when fresh and after 1, 2, 3, 4, 5 and 6 months of storage.

**Figure 4 foods-09-00791-f004:**
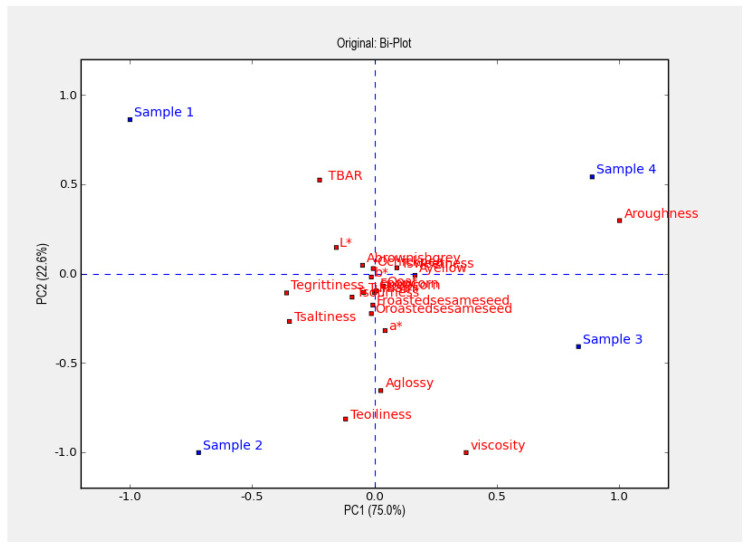
A Principal component analysis (PCA) plot of the four samples.

**Table 1 foods-09-00791-t001:** Chemical analysis of raw material, *Tenebrio molitor*, analysed by ALS Scandinavia AB.

Parameter	Mean Value (±) Standard Deviation
TS at 105 °C (%) ^1^	36.6 ± 2.0
As (mg/kg) ^2^	<0.006
Cd (mg/kg) ^2^	0.0329 ± 0.0062
Pb (mg/kg) ^2^	<0.01
Hg (mg/kg) ^2^	<0.006
Fe (mg/kg) ^2^	9.61 ± 2.09
Zn (mg/kg) ^2^	33.8 ± 7.1
Energy (kJ/100 g) ^3^	679 ± 48
Energy (kcal/100 g) ^3^	162 ± 11
Fat (g/100 g) ^3^	10.0 ± 0.50
Saturated fat (g/100 g) ^3^	2.58 ± 0.77
Monounsaturated fatty acids (g/100 g) ^3^	3.79 ± 1.14
Polyunsaturated fatty acids (g/100 g) ^3^	3.17 ± 0.95
Carbohydrate (g/100 g) ^3^	1.98 ± 0.14
Fibre (g/100 g) ^3^	0.676 ± 0.135
Protein (g/100 g) ^3^	15.8 ± 0.793
Salt (g/100 g) ^3^	0.0892 ± 0.02
Water (g/100 g) ^3^	70.5 ± 0.70
Myristic acid (C14:0), (g/100 g fat) ^4^	2.95 ± 0.88
Palmitic acid (C16:0), (g/100 g fat) ^4^	18.8 ± 5.64
Steric acid (18:0), (g/100 g fat) ^4^	3.32 ± 1.00
Oleic acid (C18:1n9c), (g/100 g fat) ^4^	36.2 ± 10.9
Linoleic acid (C18:2n6c), (g/100 g fat) ^4^	30.1 ± 9.02
Linolenic acid (C18:3n3), (g/100 g fat) ^4^	1.61 ± 0.48
Omega 3 fatty acids, total fat (g/100 g fat) ^4^	1.61 ± 0.48
Omega 6 fatty acids, total fat (g/100 g fat) ^4^	30.1 ± 9.02
Omega 3 fatty acids, total (g/100 g) ^4^	0.16 ± 0.05
Omega 6 fatty acids, total (g/100 g) ^4^	3.01 ± 0.90

^1^ Dry matter according to SS028113 (Water content by gravimetric method at 105 °C. Ash content after 550 °C). ^2^ Analysis with inductively coupled plasma sector field mass spectrometry (ICP-SFMS) according to standards: SS EN ISO 17294-1, 2 (mod) and EPA- method 200.8 (mod). ^3^ Fat content was analysed by using nuclear magnetic resonance (NMR). ^4^ The composition of fatty acids was analysed by using gas chromatography–flame ionization detection (GC-FID).

**Table 2 foods-09-00791-t002:** Retort process.

Process Data	Sample 1	Sample 4	Sample 2	Sample 3
Come-Up Time (min)	21		21	
Holding Time (min)	48		55	
Cooling Time (min)	49		52	
Total Process Time (min)	118		128	
F-value before cooling *	20	49	30	30
F-value Total *	31	56	40	40
Initial Temperature (°C)	23		23	
End Temperature (°C)	30		30	

* The F-value or the thermal death time is used for comparing heat sterilization procedures. It represents the total time–temperature combination received by the food, thus the time required to achieve a specified reduction in microbial numbers at a given temperature [25].

**Table 3 foods-09-00791-t003:** Sample code for all four samples during 6 months of storage.

Storage	Sample 1	Sample 2	Sample 3	Sample 4
Fresh	181106-1	181106-2	181106-3	181106-4
After 1 month	181206-1	181206-2	181206-3	181206-4
After 2 months	190107-1	190107-2	190107-3	190107-4
After 3 months	190206-1	190206-2	190206-3	190206-4
After 4 months	190306-1	190306-2	190306-3	190306-4
After 5 months	190405-1	190405-2	190405-3	190405-4
After 6 months	190506-1	190506-2	190506-3	190506-4

**Table 4 foods-09-00791-t004:** Sensory attributes and their definitions.

ATTRIBUTE	ABBREVIATION	DEFINITION
**Appearance**		
Roughness	Aroughness	Level of roughness of mealworm
Yellow	Ayellow	Intensity of yellow colour
Brownish-grey	Abrownishgrey	Intensity of brown-grey colour
Glossy	Aglossy	Intensity of glossiness
**Odour**		
Oat	Ooat	Odour of oats
Chickpea	Ochickpea	Odour of boiled chickpea
Roasted-Sesame-Seed	Oroastedsesameseed	Odour of roasted sesame seed
Popcorn	Opopcorn	Odour of fresh popcorn
**Taste/Flavour**		
Sweetness	Tsweetness	Basic taste sweet
Sourness	Tsourness	Basic taste sour
Saltiness	Tsaltiness	Basic taste salt
Umami	Tumami	Basic taste umami
Roasted Sesame Seed	Froastedsesameseed	Flavour of roasted sesame seed
Popcorn	Fpopcorn	Flavour of fresh popcorn
**Texture (in mouth)**		
Oiliness	Teoiliness	Fatty, oily sensation
Grittiness	Tegrittiness	Gritty perception from peel/shell

**Table 5 foods-09-00791-t005:** Mean values and standard deviations of viscosity, colour (L*, a* and b*) and TBAR.

Sample		Colour			TBAR μg/g
	Viscosity (mPas)	L*	a*	b*	
181106-1	131.2 ± 25.7	51.1 ± 0.4	4.1 ± 0.1	12.6 ± 0.2	5.08 ± 0.97
181206-1	88.0 ± 15.4	52.3 ± 0.8	3.9 ± 0.1	12.5 ± 0.4	6.81 ± 1.41
190107-1	56.8 ± 17.0	52.8 ± 1.3	3.8 ± 0.1	11.6 ± 0.3	6.13 ± 3.08
190206-1	226.4 ± 26.8	51.7 ± 0.3	4.3 ± 0.2	13.1 ± 0.9	6.29 ± 1.80
190306-1	359.2 ± 73.8	51.4 ± 1.1	4.1 ± 0.4	12.8 ± 0.7	7.53 ± 4.18
190405-1	27.2 ± 2.8	54.6 ± 0.3	3.8 ± 0.1	12.0 ± 0.2	6.25 ± 0.79
190506-1	108.0 ± 2.4	53.0 ± 0.4	3.8 ± 0.2	12.3 ± 0.5	6.17 ± 0.42
181106-2	531.2 ± 86.2	48.8 ± 0.5	4.7 ± 0.1	13.2 ± 0.5	3.48 ± 0.60
181206-2	453.6 ± 66.9	48.4 ± 0.2	4.7 ± 0.3	13.1 ± 0.7	5.04 ± 0.32
190107-2	429.6 ± 96.6	47.5 ± 1.0	4.7 ± 0.1	12.3 ± 0.5	4.12 ± 0.90
190206-2	479.2 ± 70.2	47.9 ± 0.6	5.4 ± 0.1	14.3 ± 0.5	3.44 ± 0.57
190306-2	392.8 ± 61.7	46.5 ± 0.2	4.5 ± 0.1	12.0 ± 0.4	3.96 ± 1.16
190405-2	403.2 ± 55.9	47.3 ± 0.1	4.9 ± 0.3	12.9 ± 0.7	3.48 ± 1.25
190506-2	435.2 ± 50.7	47.3 ± 0.5	4.6 ± 0.2	12.5 ± 0.4	4.40 ± 1.20
181106-3	468.0 ± 36.3	42.2 ± 1.1	5.2 ± 0.2	13.8 ± 1.0	2.32 ± 0.30
181206-3	560.8 ± 58.8	39.5 ± 1.9	4.3 ± 0.1	9.6 ± 1.0	3.20 ± 0.84
190107-3	562.4 ± 135.2	42.2 ± 0.5	5.1 ± 0.1	13.2 ± 0.5	3.80 ± 0.90
190206-3	407.2 ± 86.6	40.4 ± 0.1	5.3 ± 0.2	12.2 ± 0.3	3.44 ± 0.54
190306-3	567.2 ± 94.8	41.4 ± 0.1	4.4 ± 0.6	10.9 ± 1.3	6.09 ± 1.32
190405-3	552.8 ± 164.2	40.0 ± 1.7	4.7 ± 0.3	12.0 ± 1.6	4.88 ± 0.85
190506-3	504.0 ± 195.6	40.9 ± 0.2	4.7 ± 0.4	11.2 ± 0.7	4.60 ± 0.18
181106-4	558.4 ± 98.8	44.7 ± 1.4	4.4 ± 0.2	12.4 ± 0.9	2.76 ± 0.83
181206-4	585.6 ± 10.5	44.2 ± 0.2	4.2 ± 0.2	12.8 ± 1.4	4.56 ± 0.24
190107-4	557.6 ± 71.2	43.7 ± 0.4	3.9 ± 1.2	11.4 ± 2.2	4.48 ± 0.07
190206-4	111.2 ± 29.6	45.6 ± 1.2	4.4 ± 0.2	12.8 ± 0.4	4.24 ± 0.66
190306-4	370.4 ± 20.7	40.8 ± 0.5	4.7 ± 0.1	12.0 ± 0.6	2.64 ± 1.07
190405-4	607.2 ± 245.4	40.7 ± 1.6	4.2 ± 0.2	11.1 ± 1.4	3.76 ± 1.00
190506-4	89.6 ± 5.5	45.3 ± 0.4	4.3 ± 0.3	10.8 ± 0.8	4.84 ± 0.28

**Table 6 foods-09-00791-t006:** Significant Pearson correlations between measurements, bold text = |0.7| or larger. Bold text = significant and |0.7| or larger.

	AR	AY	ABG	AG	OO	OC	ORS	OP	TSw	TSo	TSa	TU	FRS	FP	TeO	TeG	Tb	L*	a*	b*	V
AR	x																				
AY	**0.81**	x																			
ABG			x																		
AG			0.39	x																	
OO		0.42		0.49	x																
OC						x															
ORS			0.50	0.63	0.48		x														
OP		0.38	0.54	0.47	0.57		**0.86**	x													
TSw	0.59	**0.72**			0.48		0.47	0.60	x												
TSo	**−0.70**	−0.55							−0.68	x											
Tsa	**−0.88**	−0.52	0.39						−0.47	**0.70**	x										
TU			**0.70**	0.63	0.64		**0.74**	**0.75**			0.50	x									
FRS			0.58	0.66	0.67		**0.85**	**0.84**	0.50			**0.71**	x								
FP			0.56	0.45	0.58		**0.83**	**0.97**	0.61			**0.71**	**0.9**	x							
TeO				**0.90**			0.60					0.53	0.6		x						
TeG	**−0.94**	**−0.71**	0.38						−0.47	0.62	**0.88**					x					
Tb	−0.45	−0.56		−0.44		0.42	−0.43	−0.38	−0.40							0.40	x				
L*	**−0.84**	−0.63							−0.41	0.46	0.69					**0.82**	0.64	x			
a*				0.50			0.39						0.4		0.51		−0.66	−0.50	x		
b*	−0.38									0.39	0.38					0.43			0.44	x	
V	0.44	0.46			0.41											−0.39	−0.53	**−0.72**	0.50		x

AR = ARoughness, AY = AYellow, ABG = ABrownishGrey, AG = AGlossy, OO = Ooat, OC = OChickpea, ORS = ORoastedSesameseed, OP = OPopcorn, TSw = TSweet, TSo = TSour, TSa = TSalty, TU = TUmami, FRS = FRoastedSesameseed FP = FPopcorn TeO = TeOiliness, TeG = TeGrittyness, Tb = TBAR, L* = L*, a* = a*, b* = b*, V = Viscosity, Bold text = significant and |0.7| or larger.

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
