# Peer review of "Product Quality during the Storage of Foods with Insects as an Ingredient: Impact of Particle Size, Antioxidant, Oil Content and Salt Content"

_foods, 2020, doi:10.3390/foods9060791_

Round 1

Reviewer 1 Report

  In general, this manuscript provides the information about effect of insect flour on the product quality of food model.  This paper has new information and unique study.  However, there are some problems and flaws in presentation and discussion.   I hope that my comments are very useful for the improvement of this research.

Comments

Table 1: Please use the appropriate units. For example, Cd is better the µg/kg.  I'm not sure what footnotes 3 and 4 indicate. Is it an experimental method? If so, do you analyze the Fats by NMR? Please review it.  Please indicate the abbreviation (C18:3n3?) for linolenic acid.

Experimental Design and Figure 1: Figure 1 is difficult to understand. Please consider showing experimental design (the composition of model foods) in a table.

Abbreviations in this manuscript are wrong usage, therefore, it should be checked again throughout. For example, TBARS and TBAR is mixed together and difficult to understand.

Figure 2: moths -> mouths

Figure 3: This figure is difficult to understand. Please make the vertical axis longer.  Please indicate the storage period, not the name of the sample.  Isn't there a unit for the vertical axis?

Table 5: Please explain why the viscosity values vary depending on storage periods. Please review the units of TBARS.  It is strange that the unit contains L (Litre), g is better.

I hope that my comments are very useful for the improvement of this manuscript.

Author Response

Thank you for reviewing this manuscript. All suggestions and comments from the reviewers have been taken into consideration.

Below, please find reviewers comments and the authors answers and amendments. These are given in red.

As a first comment, we would like to point out that the language in the original manuscript has been reviewed and revised by a certified language reviewer

Reviewer 1

Comments and Suggestions for Authors

  In general, this manuscript provides the information about effect of insect flour on the product quality of food model.  This paper has new information and unique study.  However, there are some problems and flaws in presentation and discussion.   I hope that my comments are very useful for the improvement of this research.

Thank you for comments and suggestions, these have been taken into account.

Comments

Table 1: Please use the appropriate units. For example, Cd is better the µg/kg.  I'm not sure what footnotes 3 and 4 indicate. Is it an experimental method? If so, do you analyze the Fats by NMR? Please review it.  Please indicate the abbreviation (C18:3n3?) for linolenic acid. .

Table 1: We want to use the same unit on all the measurements, e.g. mg/kg has been kept.

The footnotes have been clarified as following:

  1. Fat content was analysed by using NMR
  2. The composition of fatty acids was analysed by using GC-FID

Linolenic acid (C18:3n3), has been added in Table 1

Experimental Design and Figure 1: Figure 1 is difficult to understand. Please consider showing experimental design (the composition of model foods) in a table.

Figure 1 is changed and includes an overview of the design (table) and an illustration.

Abbreviations in this manuscript are wrong usage, therefore, it should be checked again throughout. For example, TBARS and TBAR is mixed together and difficult to understand.

We fully agree that it might be confusing using both TBAR and TBARs. In order to be consistent we now use the abbreviation TBAR throughout the text in the manuscript. TBARs have been replaced with TBAR values. Lines 21, 56, 239, 246, 268, 273, 309, 320.

Figure 2: moths -> mouths

Thank you for this comment, Figure 2 has now been corrected.

Figure 3: This figure is difficult to understand. Please make the vertical axis longer.  Please indicate the storage period, not the name of the sample.  Isn't there a unit for the vertical axis?

Figure 2 and figure 3 have been enlarged in order to better interpret the different attributes.

Table 5: Please explain why the viscosity values vary depending on storage periods. Please review the units of TBARS.  It is strange that the unit contains L (Litre), g is better. HENRIC LENNART - ;MAUD om colour och storage

Table 5: It is not Liter used in table 5, it is the measurement of colour, where L* corresponds to the lightness, see method description in §3.6. We have added an extra line in table 5 to clarify that L*-, a*-, and b*-values correspond to the colour measurements.

The values of TBAR have been recalculated and are in Table 5 now presented as µg/g.

Reviewer 2 Report

General questions: What kind of food is the mixture of grinded mealworms, oil, water, salt and rosemary extract? Is it a material used for making of the Swedish balls or sausages? While did you not also test only the mealworms for comparison. While did you add the oil (which is known to spoil easily) to the mealworms, which contained a lot of fat themselves. Why add water, if the mealworms contain 70% water?

22-The word „insects“ is used in the title, I suggest removing it and adding the word „storage“ instead.

30-34-The same sentences are used in abstract. Abstract should be the essence of the article. I suggest either removing the sentences from abstract completely, or at least reformulate (and shorten) the abstract version.

45 – add [9] after Chen et al, 2017

80-I am missing the information about breeding, such as temperature, humidity, developmental stage; and about feed – the feed has a high influence on the nutritional value.

89-If the Table 1 contains authors‘ own results, it should be in Results section, not in MaM section.

97-99-I don’t consider this sentence appropriate – if the consumer should fullfill the needs for iron, he would have to eat 1 kg of mealworms. I know from my own experience, that this is almost impossible.

106-107-Please explain why you used 80 g salt for finely grated mealworm samples and only 20g salt for coarsely grated samples.

Figure1-I don’t understand the meaning of the Figure 1.

122-For each of the four samples you used 4 kg of mealworms, why the total amount is 18kg?

Figures 2 and 3-The differences are difficult to see. The lines are too bold and are overlapping each other. Please consider using the column graph.

Figure 4-Similar problem. The differences are difficult to see, and the comparison of each parameter of the samples is nearly impossible. Maybe it would be more efficient to compare one parameter for both samples in one line graph, or again, use the column graph.

Author Response

Thank you for reviewing this manuscript. All suggestions and comments from the reviewers have been taken into consideration.

Below, please find reviewers comments and the authors answers and amendments. These are given in red.

As a first comment, we would like to point out that the language in the original manuscript has been reviewed and revised by a certified language reviewer

General questions: What kind of food is the mixture of grinded mealworms, oil, water, salt and rosemary extract? Is it a material used for making of the Swedish balls or sausages? While did you not also test only the mealworms for comparison. While did you add the oil (which is known to spoil easily) to the mealworms, which contained a lot of fat themselves. Why add water, if the mealworms contain 70% water?

The text in the last part of the introduction (line 68-71) has been changed, in order to make it more clear, and these line has been added/changed: In order to create model food products similar to real food oil, water, salt and antioxidant were added to finely grated or coarsely chopped mealworms. This will stress the product during storage as stability problems may occur. The model products can be compared to food products such as paté or purée, or Swedish meatballs.

Line 75-76 is slightly changed and now reads: Rosemary extract was used as antioxidant and combined…

22-The word „insects“ is used in the title, I suggest removing it and adding the word „storage“ instead.

The title is: “Product quality during the storage of foods with insects as an ingredient - impact of particle size, antioxidant, oil content and salt content”. Thus, already containing the word “storage”. Insect is the main ingredients in this food product; therefore, we think it is appropriate to have it in the title.

30-34-The same sentences are used in abstract. Abstract should be the essence of the article. I suggest either removing the sentences from abstract completely, or at least reformulate (and shorten) the abstract version.

The two first sentences in the abstract, line 12-14 have been rewritten and now read: To increase acceptability of insects as food in the Western culture, he development of attractive, high quality food products is essential. Higher acceptability of insect based food has been shown if the insects are “invisible”.

45 – add [9] after Chen et al, 2017

Thank you, this has been added.

80-I am missing the information about breeding, such as temperature, humidity, developmental stage; and about feed – the feed has a high influence on the nutritional value.

Line 85-88: Information on the rearing conditions are added: The breeding of the mealworms took place in plastic boxes, size about 300 mm * 300 mm * 100 mm in ambient temperature (approximately 22 °C). Humidity was approximately 50%. The time from egg to mealworm was 12 weeks. The mealworms were fed on oatbrans and carrot. 89

-If the Table 1 contains authors‘ own results, it should be in Results section, not in MaM section.

The analyses were performed by ALS Scandinavia AB. This has been added in the header to the table (although it was stated in the text already above the table).

97-99-I don’t consider this sentence appropriate – if the consumer should fullfill the needs for iron, he would have to eat 1 kg of mealworms. I know from my own experience, that this is almost impossible.

We are not suggesting that the daily intake of all nutrients should be covered by one meal of the food. We have added a sentence in this section in order to make this clearer, as follows: Thus, the intake of this type of food can add to the daily intake (line 106-107).

106-107-Please explain why you used 80 g salt for finely grated mealworm samples and only 20g salt for coarsely grated samples.

The background for using an experimental design is now better explained in the text, lines 68-71: In order to create model food products similar to real food oil, water, salt and antioxidant were added to finely grated or coarsely chopped mealworms. This will stress the product during storage as stability problems may occur. The model products can be compared to food products such as paté or purée, or Swedish meatballs.

Chapter 3.2 has been rewritten to better explain the design.

The design is also included in the discussion, lines 282-288.

Figure1-I don’t understand the meaning of the Figure 1.

Figure 1 is changed to better explain the experimental design.

122-For each of the four samples you used 4 kg of mealworms, why the total amount is 18kg?

Thank you for pointing out this mistake. The number is 16 instead of 18 and is corrected, now line 134.

Figures 2 and 3-The differences are difficult to see. The lines are too bold and are overlapping each other. Please consider using the column graph.

Yes we realize that it is difficult to see. Both figure 2 and figure 3 have been redrawn.

Figure 4-Similar problem. The differences are difficult to see, and the comparison of each parameter of the samples is nearly impossible. Maybe it would be more efficient to compare one parameter for both samples in one line graph, or again, use the column graph.

Figure 4 is the PCA plot, thus cannot be shown in “a column graph”. May this refers to figure 3? Both figure 2 and figure 3 have been redrawn.

Round 2

Reviewer 1 Report

Since my questions and requests were properly replied or revised, I think your manuscript have improved dramatically.

Author Response

Answer to Reviewer 1

Since my questions and requests were properly replied or revised, I think your manuscript have improved dramatically.

Thank you for comments and for re-Review.
